# Searching for Hypoglycemic Compounds from Brazilian Medicinal Plants Through UPLC-HRMS and Molecular Docking

**DOI:** 10.3390/plants14223517

**Published:** 2025-11-18

**Authors:** Elisabeth Mariano Batista, Gabriela Araújo Freire, Caio Bruno Rodrigues Martins, Raimundo Rigoberto Barbosa Xavier Filho, Gisele Silvestre da Silva, Nylane Maria Nunes de Alencar, Paulo Riceli Vasconcelos Ribeiro, Natalia Florencio Martins, Yasmim Aquino Milhome, Helcio Silva dos Santos, Marisa Jadna Silva Frederico, Luciana de Siqueira Oliveira, Kirley Marques Canuto

**Affiliations:** 1Programa de Pós-Graduação em Ciência e Tecnologia de Alimentos, Departamento de Engenharia de Alimentos, Universidade Federal do Ceará, Campus Pici, Fortaleza 60356-000, CE, Brazil; elisabethmariano@hotmail.com; 2Programa de Pós-graduação em Farmacologia, Faculdade de Medicina, Departamento de Farmacologia e Fisiologia, Núcleo de Pesquisa e Desenvolvimento de Medicamentos (NPDM), Universidade Federal do Ceará, Fortaleza 60430-275, CE, Brazil; gabrielaafreire@alu.ufc.br (G.A.F.); caiobruno@alu.ufc.br (C.B.R.M.); gihchemistry@gmail.com (G.S.d.S.); nylane@ufc.br (N.M.N.d.A.); marisafrederico@ufc.br (M.J.S.F.); 3Programa de Pós-Graduação em Ciências Naturais, Universidade Estadual do Ceará, Fortaleza 60714-903, CE, Brazil; rigobertorf@gmail.com (R.R.B.X.F.); helciodossantos@gmail.com (H.S.d.S.); 4Embrapa Agroindústria Tropical, Fortaleza 60020-181, CE, Brazil; paulo.riceli@embrapa.br (P.R.V.R.); natalia.martins@embrapa.br (N.F.M.); ymilhome@gmail.com (Y.A.M.)

**Keywords:** Acanthaceae, Fabaceae, Verbenaceae, alpha-glucosidase, disaccharidase inhibition, antidiabetic agents, diabetes

## Abstract

Type 2 diabetes mellitus remains a major global health challenge, and natural hypoglycemic compounds are of increasing interest. Aqueous extracts from five Brazilian medicinal plants (*Lippia origanoides* Kunth, *Amburana cearensis* (Allemão) A.C. Smith, *Justicia pectoralis* Jacq., *Libidibia ferrea* (Mart. ex Tul.) L.P. Queiroz, and *Spondias mombin* L.) were evaluated for α-glucosidase and intestinal disaccharidase inhibition; next, they were chemically profiled by UPLC-HRMS. Extracts of *L. origanoides*, *A. cearensis*, and *J. pectoralis* exhibited the strongest activities, attributed to flavonoids, iridoid and cinnamic acid derivatives, phenolic acids, and saponins. Molecular docking identified hyperoside, eudesmic acid and isoquercitrin as their main α-glucosidase inhibitors, respectively. In vitro testing confirmed that isoquercitrin effectively inhibited α-glucosidase (IC_50_ = 0.09 mg mL^−1^), showing stronger activity than acarbose. ADME simulations indicated low gastrointestinal absorption but favorable intestinal enzyme-targeted properties, consistent with their pharmacological profiles. Acute toxicity in zebrafish showed low toxicity for *L. origanoides* and *A. cearensis* and moderate levels for *J. pectoralis*, supporting their overall safety. These findings highlight these species as promising sources of bioactive compounds for developing safe, plant-based antidiabetic agents.

## 1. Introduction

Several medicinal plants have been widely studied as sources of bioactive compounds for treating or managing type 2 diabetes mellitus (T2DM), including *Allium sativum*, *Momordica charantia*, *Hibiscus sabdariffa* L., and *Zingiber officinale* [1]. T2DM represents one of the main public health challenges today, being characterized by insulin resistance and persistent hyperglycemia. Dysregulation of glucose metabolism has been associated with several chronic complications, such as cardiovascular diseases, nephropathy and retinopathy, in addition to contributing to systemic inflammation and prolonged oxidative stress [2].

Given the global rise in T2DM prevalence, therapeutic strategies targeting postprandial glycemia have gained increasing importance. One of the most established approaches involves inhibiting digestive enzymes responsible for carbohydrate hydrolysis, such as α-glucosidase and the intestinal disaccharidases maltase, sucrase, and lactase [3]. Under diabetic conditions, the activity of these enzymes is markedly elevated, contributing to postprandial hyperglycemia. Located on the brush border of the intestinal epithelium, they catalyze glucose release from oligosaccharides and disaccharides, thereby accelerating its absorption. Studies have shown that insulin deficiency abnormally increases disaccharidase activity in animal models [4] and that patients with T2DM exhibit approximately 1.5-fold higher α-glucosidase expression than healthy individuals [5]. This enhanced enzymatic activity intensifies intestinal glucose uptake and exacerbates metabolic imbalance, reinforcing the therapeutic relevance of α-glucosidase inhibitors such as acarbose and miglitol to control postprandial blood glucose levels [6]. However, these drugs are often associated with gastrointestinal adverse effects, including abdominal distension and flatulence, which can reduce treatment adherence [7]. Consequently, there is growing interest in identifying plant-derived bioactive compounds with α-glucosidase inhibitory activity, as they may offer effective alternatives with lower toxicity and fewer side effects. Notably, the α-glucosidase inhibitor miglitol is a semi-synthetic derivative of the iminosugar 1-deoxynojirimycin [8].

Furthermore, many classes of secondary metabolites, including flavonoids, phenolic acids, and tannins, have demonstrated the ability to inhibit α-glucosidase and disaccharidases, besides exhibiting other pharmacological activities such as antioxidant and anti-inflammatory actions [6,9]. Despite numerous reports describing enzyme inhibition by plant-derived compounds, most studies have focused on species traditionally used for diabetes treatment. However, several tropical plants with well-documented medicinal properties remain chemically and pharmacologically underexplored for this purpose. We hypothesize that their flavonoid- and phenolic acid-rich extracts may be interesting sources of antidiabetic compounds by modulating carbohydrate metabolism through enzyme inhibition and complementary antioxidant mechanisms.

Therefore, in the present work, we have screened extracts from five Brazilian medicinal plants, namely, *Amburana cearensis (Allemão) A.C. Smith*, *Justicia pectoralis* Jacq., *Lippia origanoides* Kunth, *Libidea ferrea* (Mart. ex Tul.) L.P. Queiroz and *Spondias mombin* L.), against α-glucosidases; next, their constituents were characterized by Ultra-Performance Liquid Chromatography coupled with High-Resolution Mass Spectrometry (UPLC-HRMS) and analyzed by molecular docking to identify the possible bioactive compounds. UPLC-HRMS has emerged as a key analytical platform for metabolomic approaches aiming at the rapid characterization of bioactive compounds in plant extracts. UPLC-HRMS combines efficient chromatographic separation with high mass accuracy, enabling the simultaneous identification of multiple metabolites in complex plant matrices. The exact mass data allow reliable determination of molecular formulas and dereplication of known bioactive compounds. By generating detailed chemical fingerprints, this technique accelerates phytochemical screening and identification of biologically relevant metabolites, promoting a more efficient exploration of natural products [10].

All species studied are dicotyledons that occur across many countries of South and Central America, except *L. ferrea*, which is endemic to Brazil. Both *A. cearensis* (Fabaceae) and *Justicia pectoralis* (Acanthaceae) have been used for treating respiratory affections (e.g., cough, asthma and bronchitis), and their anti-inflammatory and bronchodilator activities have been attributed to coumarins, phenolic acids and flavonoids [11,12]. *L. origanoides* (syn.: *L. sidoides*, Verbenaceae) has been reported for the control of bacterial and fungal infections due to its thymol-rich essential oil [13], while *L. ferrea* and *S. mombin* (Anacardiaceae) possesses antimicrobial and wound-healing properties owing to tannins [14]. Thus, the choice of the medicinal plants studied was chiefly driven by their previously described chemical constituents, instead of the record of ethnopharmacological uses for diabetes.

## 2. Results

### 2.1. Screening for α-Glycosidase Inhibition Activity of Extracts from Brazilian Medicinal Plants

As shown in Table 1, the aqueous extract of *L. origanoides* leaves showed the greatest inhibitory capacity on the enzyme α-glucosidase (IC_50_ of 0.485 mg/mL), followed by the extracts of *J. pectoralis* and *A. cearensis*. Although their values of IC_50_ have been higher than acarbose (positive control), the result reveal a great potential inhibitory for these extracts, taking into account that these samples are a complex mixture of several components (Appendix A). On the other hand, the extracts of *L. ferrea* and *S. mombin* did not show activity.

Afterwards, the three aforementioned bioactive extracts were assessed for their inhibitory activity against intestinal disaccharidases, exhibiting distinct profiles. According to Figure 1a, *A. cearensis* extract inhibited the sucrase enzyme by approximately 85% and 79% at concentrations of 0.312 mg/mL and 10 mg/mL, respectively. *L. origanoides* extract reduced sucrase activity by about 86% and 63%, while *J. pectoralis* demonstrated inhibition rates of 84% and 87% at the same concentrations. Acarbose, used as a standard inhibitor, suppressed sucrase activity by approximately 71%.

In the lactase assay (Figure 1b), the extract of *A. cearensis* also led to a marked decrease in enzymatic function, with reductions of around 77% and 70% at 0.312 mg/mL and 10 mg/mL, respectively. *L. origanoides* extract inhibited lactase by 78% and 50%, whereas *J. pectoralis* showed inhibition of approximately 74% and 80%. Acarbose reduced lactase activity by about 51% under the same conditions. Regarding maltase assay (Figure 1c), *A. cearensis* extract promoted a 59% reduction in activity at 10 mg/mL, whereas *L. origanoides* and *J. pectoralis* did not present significant inhibition at either tested concentration. In comparison, acarbose inhibited maltase activity by nearly 74%.

### 2.2. Toxicological Evaluation of the Bioactive Extracts

The toxicological assessment of the bioactive extracts in adult zebrafish (*Danio rerio*) indicated low acute toxicity for *L. origanoides* and *A. cearensis* extracts, with negligible mortality observed up to the highest tested concentration (100 mg mL^−1^) (Appendix A). The LC_50_ values for both extracts were estimated to exceed this limit, supporting their classification as low-hazard substances according to the OECD Test Guideline 203 and GHS/CLP criteria. In contrast, *J. pectoralis* extract produced limited mortality at 50 mg mL^−1^, yielding an estimated LC_50_ above this concentration but below 100 mg mL^−1^. Consequently, it can be provisionally classified within GHS Category 3 (LC_50_ between 10 and 100 mg mL^−1^), pending further assays to refine its toxicological profile.

### 2.3. Chemical Characterization of the Bioactive Extracts

UPLC-HRMS chromatograms (Figure 2) exhibited 21, 28 and 15 peaks in the extracts of *A. cearensis*, *J. pectoralis* and *L. origanoides*, respectively, of which 16, 21 and 14 were tentatively identified. According to Table 2, *L. origanoides* extract consisted of nine flavonoids (**2**, **24**, **37**, **38**, **44**, **46** and **49–51**), three iridoids (**3**, **8** and **31**), two phenylpropanoids (**22** and **33**), and two carboxylic acids (**1** and **16**). Shanzhiside (**3**), lippioside I (**31**), verbascoside (**33**), catechin-*O*-hexoside (**37**) and kaempferol-*O*-acetylhexoside (**46**) presented the most intense peaks.

*A. cearensis* extract consisted of eight phenolic acids (**4**, **10**, **29**, **34**, **40**, **43**, **47** and **48**), six flavonoids (**12**, **14, 17**, **20**, **25** and **41**), two monosaccharide derivatives (**5** and **9**), three triterpenoidal saponins (**54**, **55** and **58**) along with eight unidentified saponins (**56**, **57** and **59–64**). Amburoside A (**34**), amburoside B (**43**) and ferulic acid (**48**) were found to be the most abundant compounds. The saponins soysaponins Be (**56**) and Bg (**57**) as well as hydroxyoleanenyl-deoxy-mannopyranosyl-hexopyranosyl-glucopyranosiduronic acid (**58**) have been reported for the first time in *A. cearensis*.

In *J. pectoralis* extracts, we found seven hydrolyzable tannins (**11**, **13**, **15**, **18**, **23**, **26** and **27**), three ellagic acid derivatives (**21**, **30** and **36**), two flavonoids (**32** and **35**), one phenylpropapoid (**7**) and one flavolignan (**39**). Chlorogenic acid (**7**), corilagin (**13**) and chebulagic acid (**23**) were the main components.

### 2.4. Molecular Docking of the Bioactive Compounds and Their ADME Properties

Molecular docking is a computational method used to predict the interaction between small molecules and biological targets. It helps to identify potential drug candidates by estimating binding affinity and orientation within the active site. This technique enables virtual screening of large compound libraries, guiding experimental efforts. It also aids in lead optimization and understanding mechanisms of action [45].

Based on the molecular docking results shown in Table 3, hyperoside (quercetin-3-*O*-galactose) demonstrated the most promising binding potential with a score of −7.994 kcal/mol, slightly outperforming the positive control acarbose, which scored −7.465 kcal/mol. Likewise, its isomer isoquercitrin (quercetin-3-*O*-glucose) showed strong binding affinity with a score of −7.514 kcal/mol. In contrast, eudesmic acid (trimethoxy-benzoic acid) exhibited the weakest binding affinity at −6.391 kcal/mol. Visualization of the complex in ChimeraX confirmed the quercetin core is well-positioned in the catalytic center, forming direct contacts with active site residues and demonstrating high spatial complementarity (Figure 3). In contrast, the trimethoxy-benzoic acid performed poorly (Appendix A). Our findings are consistent with previous studies that reported a superior inhibition of quercetin glycosides over acarbose, highlighting the potential of these natural flavonoids as alternative α-glucosidase inhibitors for type 2 diabetes treatment [46,47].

The ADME and drug-likeness predictions support the pharmacological interpretation of the docking results by highlighting differences in molecular properties that influence absorption and systemic availability. Hyperoside (quercetin-3-*O*-galactoside) and isoquercitrin (quercetin-3-O-glucoside) displayed physicochemical profiles typical of natural product glycosides, including high polarity and an elevated number of hydrogen bond donors and acceptors. These characteristics led to multiple Lipinski’s rule of five violations, resulting in low predicted gastrointestinal (GI) absorption and poor blood–brain barrier permeability. Such profiles are consistent with their limited oral bioavailability but may still allow local intestinal enzyme inhibition, which aligns with their observed α-glucosidase and disaccharidase inhibitory effects. In contrast, trimethoxybenzoic acid exhibited a favorable drug-likeness profile, showing no Lipinski’s violations, good oral bioavailability (bioavailability score = 0.85), high predicted water solubility, and potential blood–brain barrier permeability. These properties suggest that this compound could achieve systemic exposure and potentially exert both central and peripheral pharmacological effects. Meanwhile, acarbose—although a clinically established α-glucosidase inhibitor—violates three of Lipinski’s rules, consistent with its poor intestinal absorption and low oral bioavailability. Despite this, its therapeutic efficacy relies on localized gastrointestinal action, which parallels the mechanism proposed for quercetin glycosides. Importantly, none of the analyzed compounds were predicted to exhibit cytochrome P450 inhibitory liabilities, minimizing the likelihood of drug–drug interactions. However, their relatively high synthetic accessibility scores, particularly for acarbose, indicate substantial structural complexity (see Appendix A).

Lastly, to validate the molecular docking predictions, an in vitro α-glucosidase inhibition assay was conducted using an authentic analytical standard of isoquercitrin. This flavonoid glycoside exhibited an IC_50_ value of 0.09 ± 0.004 mg/mL (194 µM), which was actually lower than that of acarbose (IC_50_ = 0.159 mg/mL or 247 µM), indicating a higher inhibitory potency. The corresponding dose–response curve for isoquercitrin can be found in the Appendix A.

## 3. Discussion

Intestinal α-glucosidases are key enzymatic complexes that catalyze the hydrolysis of dietary oligosaccharides and polysaccharides into α-D-glucose monomers, enabling efficient carbohydrate absorption. In diabetes, α-glucosidase activity is abnormally elevated due to impaired glucose uptake by muscle and adipose tissues. This metabolic imbalance contributes to postprandial hyperglycemia, a major risk factor for cardiovascular complications. Therefore, inhibition of α-glucosidases has emerged as a validated therapeutic target, as it delays carbohydrate digestion and reduces glucose absorption rates. α-Glucosidase inhibitory assays have been widely employed for screening candidate hypoglycemic compounds, elucidating their mechanisms of action, and supporting biomarker discovery for diabetes management [2,47,48].

In our study, five Brazilian medicinal plants (*A. cearensis*, *L. origanoides*, *J. pectoralis*, *L. ferrea* and *S. mombin*) were screened for in vitro and ex vivo α-glucosidase activities. Since the aqueous extracts of *L. origanoides*, *A. cearensis* and *J. pectoralis* leaves were found to be the most bioactive, they were assessed for acute toxicity and chemically characterized, and then their components were assessed by molecular docking for identifying the α-glucosidase inhibitor compounds.

The aqueous extract of *L. origanoides* presented the highest hypoglycemic potential by inhibiting the α-glucosidase enzyme and reducing the activity of intestinal dissacharidases. Earlier, Miranda et al. [49] found a remarkable α-glucosidase inhibitory activity (94–98%), using hydroalcoholic extract of the aerial parts of *L. origanoides* at the concentrations of 5 and 10 mg/mL. Furthermore, this same extract (250 mg/kg) reduced hyperglycemia in alloxan-induced diabetic rats. The authors attributed the hypoglycemic effect to the synergistic action of many flavonoids that had previous report of lowering the blood glucose such as naringenin, apigenin, vicenin, cirsimaritin, orientin, sakuranetin, aromadendrin, vitexin, and isorhamnetin-3-*O*-rutinoside, using different mechanism of action including α-glucosidase inhibition. Likewise, our extract was rich in flavonoids; however, only naringenin (**50**) was present in both extracts. This difference in chemical composition between the extracts might be due to part plant used and the extraction method (e.g., solvent type). In fact, naringenin (**50**), which is one of the major flavonoids found in citrus fruits (grapefruit and orange), demonstrated to inhibit in vitro and in vivo α-glucosidase activities, leading to significant lowering of postprandial blood glucose levels [50].

Additionally, previous studies have described α-glucosidase inhibitory activity for other compounds identified from *L. origanoides* extract: secologanoside (**8**)] [51], caffeic acid (**22**) [52], verbascoside (**33**) [53], which is one of the major compounds, besides kaempferol (**49**) and its derivatives (**46** and **51**) [54] as well as quercetin glycosides such as quercetin-3-*O*-galactose (hyperoside, **24**) and quercetin acetylhexoside (**38**) [47]. Incidentally, among the constituents of the *L. origanoides* extract, hyperoside (**24**) exhibited the strongest predicted binding affinity to α-glucosidase in the molecular docking assays. Our result is in agreement with those from literature [47].

With regard to *A. cearensis* and *J. secunda*, both extracts exhibited remarkably α-glucosidase inhibitory activity. Notably, the former was the only extract capable of inhibiting all intestinal disaccharidases, while the latter showed the second strongest inhibition of α-glucosidase among the five plants extracts assayed. Although there are no studies reporting the potential hypoglycemic of *A. cearensis* and *J. pectoralis*, several works have described the effects of many of their constituents identified herein.

In *A. cearensis*, kaempferol and quercetin glycosides (**12**, **14, 17**, **20** and **25**) [54], phenolic acids such as vanillic acid hexoside (**4**), caffeic glucoronide (**10**), coumaric acid (**47)** and ferulic acid (**48**) [52], and soysaponins Be (**56**) and Bg (**57**) [43] have demonstrated inhibitory activity on α-glucosidase. Molecular docking revealed eudesmic acid (trimethoxy-benzoic acid, **29**) as the best α-glucosidase ligand for *A. cearensis* extract. Also, Chike-Ekwughe et al. [55] found eudesmic acid as one of top-ranking α-glucosidase ligands among the constituents from *Tapinanthus cordifolius* leaf extract. On the other hand, Sancheti et al. [56] did not find any activity for this compound in α-glucosidase assays.

Among the components from *J. pectoralis*, previous reports showed α-glucosidase inhibitory activity for chlorogenic acid (**7**) [57], the hydrolyzable tannins geraniin (**11**), corilagin (**13**) and chebulagic acid (**23**) [58], the phenolic acids ellagic acid (**30**) and methyl-ellagic acid (**36**), and the flavonoids kaempferol and quercetin glycosides (**32** and **35**) [54]. Geraniin (**11**) is an ellagitannin with well-documented antidiabetic properties through in vitro and pre-clinical studies. Since geraniin has a high molecular weight and strong polarity, its direct absorption in the intestinal tract is limited. Instead, it is metabolized by gut microbiota into smaller compounds such as corilagin (**13**), ellagic acid (**30**) and gallic acid, which are more bioavailable. Hence, these metabolites, rather than geraniin itself, are primarily responsible for their observed antidiabetic effects [59]. In our in silico evaluation, isoquercitrin (**32**) was predicted as best binding to α-glucosidase, corroborating the result found by Abudurexiti et al. [46] investigating α-glucosidase inhibitors from mulberry. Furthermore, isoquercitrin exhibited strong in vitro α-glucosidase inhibitory activity, with an IC_50_ value of 0.09 ± 0.004 mg/mL (194 µM), demonstrating greater potency than acarbose (IC_50_ = 0.159 mg/mL; 247 µM). These findings are consistent with those reported by Seong et al. (2023) [60], who observed IC_50_ values of 87 µM for isoquercitrin and 270 µM for acarbose, respectively. Thus, these experimental results strongly corroborate the in silico findings, supporting the hypothesis that isoquercitrin is the main active constituent responsible for the α-glucosidase inhibitory activity observed in the *Justicia pectoralis* extract.

With relation to safety, *L. origanoides* and *A. cearensis* extracts showed negligible acute toxicity in zebrafish, while *J. pectoralis* presented low to moderate toxicity (GHS Category 3). These results indicate overall safety for pharmacological use, though further studies are needed to better characterize the toxicological profile of *J. pectoralis.*

Regarding the non-active plant extracts *Libidibia ferrea* and *Spondias mombin*, only the latter has previously been reported to exhibit antidiabetic properties. In Southwestern Nigeria, traditional healers have used *S. mombin* leaves to manage diabetes. Methanolic extracts of *S. mombin* leaves, at doses of 125 and 1000 mg/kg, were shown to reduce blood glucose levels in streptozotocin- and alloxan-induced diabetic rats, respectively. However, since the chemical composition was not characterized in either study, no specific bioactive compounds could be linked to the observed hypoglycemic effects.

## 4. Material and Methods

### 4.1. Chemicals

LC-MS-grade acetonitrile was purchased from Tedia Co. (Fairfield, CT, USA) and formic acid was obtained from Fluka Co. (Buchs, ZU, Switzerland). Ultrapure water was provided by a Milli-Q water purification system (Billerica, MA, USA). Analytical standards (quinic acid, narigenin, quercetin, isoquercitrin, chlorogenic acid, geraniin, corilagin, caffeic acid, ellagic acid, ferulic acid and kaempferol), besides the pharmaceutical reagents acarbose (>95%), nitrophenyl α-D-glucopyranoside (>99%) and the enzyme α-glucosidase (≥10 units/mg, sourced from Saccharomyces cerevisiae), were purchased from Sigma-Aldrich (Saint Louis, MO, USA). Amburosides A and B were previously isolated from *A. cearensis* seedlings by Canuto et al. [35]. All other reagents used were prepared from chemicals of analytical grade.

### 4.2. Plant Material and Preparation of Plant Extracts

The leaves of *Lippia origanoides*, *Justicia pectoralis*, *Amburana cearensis *(Figure 4), *Spondias mombin*, and *Libidibia ferrea* were collected at the Medicinal Plant Garden Professor Francisco José de Abreu Matos of the Federal University of Ceará in the city of Fortaleza, Ceará, Brazil. These plants were authenticated by a botanist and their exsiccates were deposited at Prisco Bezerra Herbarium under the following voucher specimen numbers: #60230, #61198, #68159, #65551 and #58169. The research on these plants was registered in SisGen (National System for the Management of Genetic Heritage and Associated Traditional Knowledge) under number A9A36B5, as required by Brazilian legislation. Briefly, after collection and transport, the leaves were cleaned with distilled water, dried in an oven at 40 °C, ground with the help of an industrial blender, and stored in zip-lock laminated bags in a desiccator until extraction. The aqueous extracts were obtained by infusion with distilled water at 100 °C, in a 1:40 (*w*/*v*) ratio, for 5 min according to Brazilian Pharmacopeia (ANVISA, 2021). After filtration, the extracts were lyophilized and stored in zip-lock laminated bags at −18 °C until analysis.

**Figure 4 plants-14-03517-f004:**
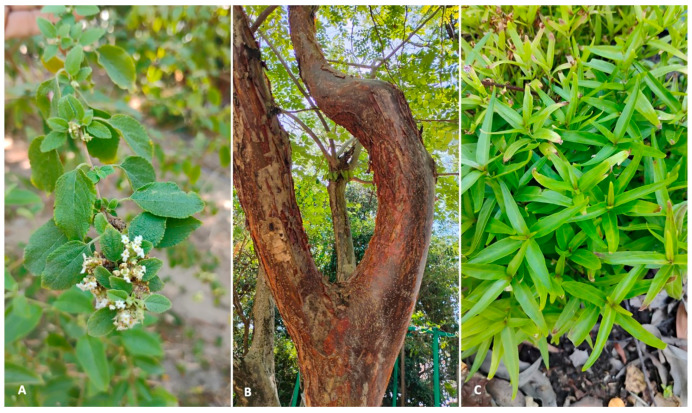
Photographs of the plants with potential antidiabetic properties: (**A**)—*Lippia origanoides*; (**B**)—*Amburana cearensis*, (**C**)*—Justicia pectoralis*.

### 4.3. In Vitro α-Glucosidase Inhibition Assay

The assessment of α-glucosidase inhibitory activity was conducted following the methodology adapted from Vinholes et al. [61], with specific adjustments [62]. Initially, 100 μL of various sample concentrations, previously dissolved in 0.1 M phosphate-buffer solution at pH 6.8, were distributed in triplicate into a microplate. Afterward, 50 μL of glutathione was added to each well containing the samples. Subsequently, 50 μL of the α-glucosidase enzyme, diluted to a final concentration of 0.4 U/mL in 0.1 M phosphate-buffer solution, was added. The microplates were incubated at 37 °C for 5 min, and the initial absorbance value (T0’) was recorded at 415 nm using an Eon BioTek spectrophotometer (Winooski, VT, USA). The enzymatic reaction was activated by adding 50 μL of substrate (4-nitrophenyl α-D-glucopyranoside at 10 mM in 0.1 M phosphate-buffer solution) and incubated in the dark for 10 min (T10’). At the end of this period, a new absorbance reading was taken at 415 nm. Acarbose, solubilized in 0.1 M phosphate-buffer solution at different concentrations, served as the positive control of this assay. Also, an absorbance reading for the negative control was performed using phosphate buffer in place of the sample. The inhibitory activity was calculated using the following equation: Inhibitory activity % = [(Control − Sample)/(Control)] × 100. The sample concentration required to inhibit enzyme activity by 50% (IC50) was calculated by nonlinear regression analysis.

### 4.4. Animals

Male C57BL/6 mice (7 weeks, 10–20 g) were kept in cages at 21 ± 2 °C and a light-dark cycle of 12 h (light on between 6 and 6 h) with feed (Nuvital, Curitiba, PR, Brazil) and water ad libitum. The experiments were carried out in the bioterium of the Center for Research and Development of Medicines (NPDM) at Federal University of Ceará according to the Ethics Committee on Animal Use (CEUA-NPDM/UFC: N° 09150322-0). This bioterium is certified by Association for Assessment and Accreditation of Laboratory Animals (AAALAC).

For toxicity assay, adult wild-type zebrafish (*Danio rerio*) of both sexes were obtained from a local supplier (Agroquímica Produtos Veterinários, Fortaleza, CE, Brazil). The specimens were approximately 60–90 days old, measuring 3.5 ± 0.5 cm in length and weighing 0.4 ± 0.1 g. Upon arrival, fish were acclimated for 24 h in rectangular glass tanks (30 × 15 × 20 cm) containing dechlorinated water (ProtecPlus^®^) maintained at 25 °C and neutral pH, and equipped with air pumps and submerged filters, as described by Magalhães et al. [63]. Zebrafish were maintained under a 14:10 h light/dark photoperiod and fed *ad libitum* until 24 h prior to experimentation. All experimental procedures were approved by the Animal Ethics Committee of the Federal University of Ceará (CEUA-UFC; protocol no. 2101202201/2025).

### 4.5. Ex Vivo Intestinal Disaccharidases Inhibition Assay

The activity of disaccharidases, including maltase, sucrase, and lactase, was determined following the methodology described by Dahlqvist with modifications [64,65]. The mouse’s duodenum was removed, washed in 0.9% NaCl solution, cut into small fragments (2.0 cm each), incubated with 300 µL of maleate buffer and treated with extracts of *J. pectoralis*, *A. cearensis* and *L. origanoides* at concentrations of 0.312 and 10 µM for 20 min. Acarbose was used as a positive control and assessed at concentrations of 500 μM (0.32 mg/mL), 700 μM (0.45 mg/mL), and 900 μM (0.58 mg/mL), with the intermediate concentration being used as a reference. The concentrations were selected based on an alpha-glucosidase experiment, using only the minimum and maximum, given the experimental limitations. Subsequently, the supernatant was used for ex vivo evaluation of the α-glucosidase activity and determination of total proteins. The samples were read at 500 nm using a spectrophotometer. The values were expressed as enzyme activity (U) per milligram of protein. Additionally, values of IC_50_ were estimated based on the aforementioned doses used for the sample and positive control.

### 4.6. Zebrafish Acute Toxicity Assay

For experimental procedures, fish were randomly assigned to treatment groups, gently immobilized on a moist sponge, and orally administered 20 µL of either the plant extract or vehicle (control) using a micropipette fitted with a flexible tip, following the protocol of [66]. Extract concentrations were determined based on previously established minimum bactericidal concentrations. Acute toxicity was assessed over 96 h in groups of zebrafish (n = 6 per group). Each fish received a single oral dose of 20 µL of either the extract or the control vehicle. Following administration, animals were monitored for 96 h to record mortality and behavioral alterations. Mortality data were used to calculate the median lethal concentration (LC_50_), defined as the concentration causing 50% mortality among exposed organisms, as described by [67]. At the end of the assay, fish were euthanized by immersion in ice-cold water (2–4 °C) for 10 min, until opercular movements ceased completely.

### 4.7. Statistical Analyses

All data were analyzed using GraphPad Prism 9 (GraphPad Software, La Jolla, CA, USA). The normality of the data was assessed using the Shapiro–Wilk test for each experimental group. Data were considered normally distributed when *p*-values were greater than 0.05. Additionally, the Kolmogorov–Smirnov test was applied when necessary to further confirm the results from the Shapiro–Wilk test. To evaluate the homogeneity of variances among groups, the Brown-Forsythe test was used, which is robust even when data do not follow a normal distribution. Homogeneity of variances was considered met when the *p*-value was greater than 0.05. A one-way analysis of variance (ANOVA) was conducted to compare the effects of treatments relative to controls. When appropriate, Dunnett’s post hoc test was used for multiple comparisons between groups, with differences considered significant at *p* < 0.05 (Appendix A). Statistical significance was set at α = 0.05 for all analyses. Results are presented as means ± Standard Error of the Mean (SEM) from two independent experiments (n = 2), with each experiment performed in technical triplicates (n = 3).

### 4.8. UPLC-HRMS Analyses

The analyses were performed using an Acquity UPLC (Waters, Milford, MA, USA) system coupled to a mass system (Q-TOF, Waters). A Waters Acquity BEH C18 column for separation condition (150 mm × 2.1 mm, 1.7 μm) was set in 40 °C. An injection volume of 5-μL aliquot of each extract was subjected to an exploratory gradient with the mobile phase composed of deionized water (A) and acetonitrile (B), both containing formic acid (0.1% *v*/*v*). The extracts were subjected to an exploratory gradient as follows: 2–100% B (22.0 min), 100% B (22.1–25.0 min), 2% B (26.0–30.0 min) with flow rate of 0.3 mL/min. Ionization was performed using an electrospray ionization source in positive mode ESI mode, acquired in the range of 110–1180 Da and the optimized instrumental parameters were as follows: capillary voltage at 3.2 kV, cone voltage at 15 V, source temperature at 120 °C, desolvation temperature at 350 °C, desolvation gas flow at 500 L/h. The system was controlled using Masslynx software (Waters Co.). The compounds were tentatively identified through molecular formula provided by MassLynx 4.1 software based on their accurate masses (error < 5 ppm), isotopic patterns (i-fit) and MS fragmentation patterns, besides chemotaxonomic survey using SciFinder database. Additionally, compounds were identified by comparison with reference standards when available.

### 4.9. Molecular Docking Studies

Molecular docking studies were conducted to evaluate the binding of compounds from bioactive extracts to α-glucosidase (PDB ID: 2QMJ). The protocol began with protein and ligand preparation. The 3D structure of the target enzyme was obtained from the Protein Data Bank. A ligand library was constructed using compounds identified in the extracts; their PubChem CID numbers were used to download the corresponding 3D structures in .sdf format from the PubChem database. For validation, the co-crystallized ligand was removed and acarbose—a known inhibitor used as a reference—was obtained in the same format for redocking. The docking simulations were performed using the DockThor software v 2.0 [68], a platform recognized for its accuracy in screening natural products. The Root Mean Square Deviation (RMSD) between the crystallographic and re-docked conformations was 1.9 Å, confirming that the docking protocol accurately reproduced the experimental binding pose (RMSD ≤ 2.0 Å). The α-glucosidase catalytic pocket in 2QMJ corresponds to a deep cleft within the (β/α)_8_ TIM barrel, containing two key catalytic aspartate residues and surrounded by aromatic side chains that stabilize ligand stacking. To encompass this region, the grid box was centered at coordinates x = 55.41 Å, y = 98.61 Å, and z = 20.59 Å, with a grid spacing of 0.42 Å, ensuring inclusion of the catalytic residues and adjacent subpockets relevant for inhibitor binding.

Following execution, the resulting poses were ranked by their binding affinity energy (kcal/mol). The top-ranked compound based on affinity was selected for further analysis. Its structure was superimposed onto the acarbose pose within the enzyme’s active site using ChimeraX software v.1.10 [69] to visualize its binding mode. Finally, the most promising protein–ligand complexes were analyzed in detail using LIGPLOT+ v 2.3 [70] to map specific molecular interactions, including hydrogen bonds and hydrophobic contacts. Furthermore, pharmacokinetic profiles—encompassing Absorption, Distribution, Metabolism, and Excretion (ADME)—were predicted using the SwissADME web tool (http://www.swissadme.ch). The smiles files were uploaded directly to the server [71]. The analyses were run using the default parameters for all compounds.

## 5. Conclusions

In summary, this study demonstrated that aqueous extracts from *Lippia origanoides*, *Justicia pectoralis and Amburana cearenses* exhibit promising α-glucosidase and disaccharidase inhibitory activities, reinforcing their potential as phytotherapeutic candidates for the management of type 2 diabetes mellitus. Furthermore, toxicological evaluation in adult zebrafish revealed low acute toxicity for *L. origanoides* and *A. cearensis*, with estimated LC_50_ values above 100 mg mL^−1^, whereas *J. pectoralis* induced limited mortality (LC_50_ > 50 mg mL^−1^), being provisionally classified within GHS Category 3.

*L. origanoides* extract displayed the highest inhibitory potential, which could be attributed primarily to a diverse array of flavonoids—including naringenin, kaempferol, and quercetin and their glycosides—cinnamic acid derivatives (caffeic acid and verbascoside), along with the iridoid secologanoside. Molecular docking identified hyperoside as a compound with strong predicted affinity for α-glucosidase, comparable to acarbose, suggesting its potential contribution to the observed inhibitory activity.

*Justicia pectoralis* extract exhibited substantial activity against both α-glucosidase and disaccharidases, associated with hydrolyzable tannins (e.g., geraniin, corilagin, chebulagic acid), phenylpropanoids (chlorogenic acid), ellagic acid derivatives, and flavonoids such as kaempferol and isoquercitrin. Docking simulations identified isoquercitrin as the most promising ligand in this extract, corroborating prior evidence of its antidiabetic potential, which was further confirmed experimentally through in vitro α-glucosidase inhibition (IC_50_ = 0.09 mg/mL or 194 µM), showing higher potency than acarbose. Furthermore, ADME predictions revealed that isoquercitrin and its isomer hyperoside display limited gastrointestinal absorption yet remain pharmacologically relevant due to their suitability for local intestinal enzyme inhibition.

*Amburana cearensis* extract also showed relevant inhibitory activity, particularly notable for its broad-spectrum inhibition of intestinal disaccharidases (sucrase, lactase, and maltase). This effect is likely mediated by the presence of phenolic acids (ferulic acid, vanillic acid hexoside, and caffeic acid glucuronide), flavonoid glycosides (kaempferol and quercetin derivatives), and triterpenoid saponins such as soyasaponins Be and Bg. Eudesmic acid (trimethoxybenzoic acid), although less studied, exhibited favorable docking interactions and ADME parameters.

Altogether, these findings support the pharmacological relevance and safety of these Brazilian medicinal plants as sources of α-glucosidase inhibitors. The integration of enzyme inhibition assays, phytochemical profiling, toxicological testing, and in silico analysis provided robust evidence to guide future in vivo and clinical investigations aimed at the development of safe, plant-based hypoglycemic agents.

## Figures and Tables

**Figure 1 plants-14-03517-f001:**
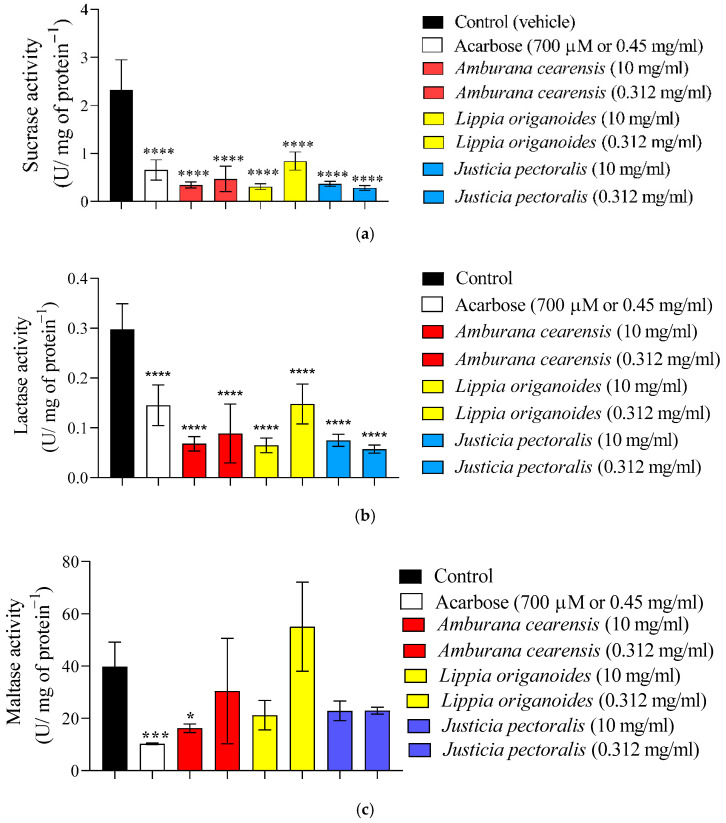
Effect of aqueous extracts of *L. origanoides*, *J. pectoralis* and *A. cearensis* along with acarbose on ex vivo disaccharidases enzymes assays: (**a**) Inhibition on sucrase activity; (**b**) Inhibition on lactase activity and (**c**) Inhibition on maltase activity. Values are expressed as mean ± S.E.M.; *n* = 2; performed in triplicate. **** *p* ≤ 0.0001; *** *p* ≤ 0.001 and * *p* ≤ 0.05, compared to the control group.

**Figure 2 plants-14-03517-f002:**
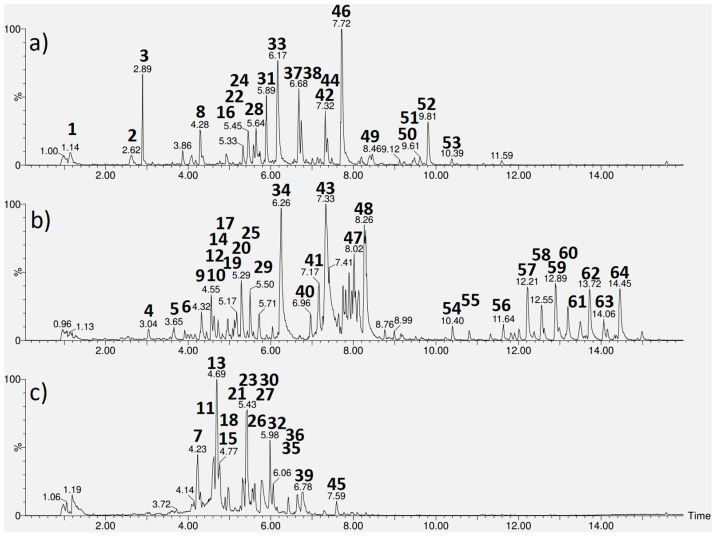
UPLC-HRMS chromatograms from the aqueous extracts of the leaves from *Lippia origanoides* (**a**), *Amburana cearensis* (**b**) and *Justicia pectoralis* (**c**). The peak numbers corresponding to the components identified following the elution order based on their retention times.

**Figure 3 plants-14-03517-f003:**
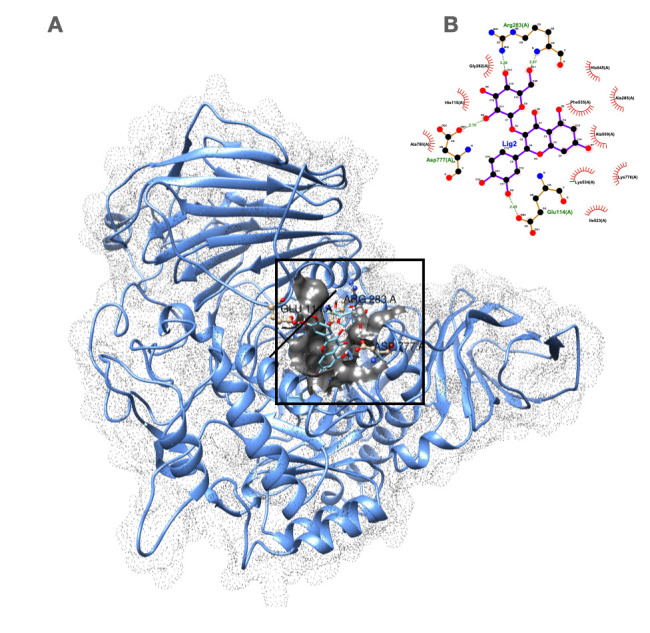
Molecular interaction of hyperoside (quercetin-3-*O*-galactose) with the human α-glucosidase active site. (**A**) Three-dimensional representation of the docked pose. The protein surface is shown as a transparent dot field. Hyperoside is depicted as ball-and-stick. Residues forming hydrogen bonds are shown as ball-and-stick and labeled; residues involved in hydrophobic interactions are highlighted with a gray solid surface. (**B**) LIGPLOT two-dimensional ligand-protein interaction diagram. Hydrogen bonds are shown as green dashed lines, and hydrophobic contacts are indicated by red arcs.

**Table 1 plants-14-03517-t001:** Inhibitory activity of aqueous extracts of leaves from five Brazilian medicinal plants against α-glucosidase enzyme measured as half-maximal inhibitory concentration (IC_50_).

Extracts	Inhibition	IC_50_ (mg/mL)
*Lippia origanoides*	Yes	0.485 ± 0.0851
*Justicia pectoralis*	Yes	0.812 ± 0.0486
*Amburana cearensis*	Yes	1.579 ± 0.478
*Libidibia ferrea*	No	-
*Spondias mombin*	No	-
Acarbose (positive control)	Yes	0.159 ± 0.0059

Values expressed as mean ± SEM; *n* = 2; performed in triplicate.

**Table 2 plants-14-03517-t002:** UPLC-HRMS- based chemical characterization of the extracts of *Lippia origanoides* (A), *Amburana cearensis* (B) and *Justicia pectoralis* (C).

Peak ^a^	Rt Min.	[M − H]^−^ Observed	[M − H]^−^ Calculated	MS/MS Fragments ^b^	Empirical Formula	Ppm (Error)	Putative Name	Extract	Reference
1	1.14	195.0502	195.0505	-	C_6_H_11_O_7_	−1.5	Quinic acid ^#^	A	[15]
2	2.62	467.1188	467.1190	**305**	C_21_H_23_O_12_	−0.4	Epigallocatechin-*O*-hexoside	A	[16]
3	2.89	391.1230	391.1240	**229**, 211, 185, 167	C_16_H_23_O_11_	−2.6	Shanzhiside	A	[17,18]
4	3.04	329.0883	329.0873	**167**	C_14_H_17_O_9_	3.0	Vanilic acid hexoside	B	[19]
5	3.65	209.0300	209.0297	**191**	C_6_H_9_O_8_	1.4	Mucic acid isomer	B	[20]
6	3.86	481.1338	481.1346	319	C_22_H_25_O_12_	−1.7	Unknown	A	-
7	4.23	353.0871	353.0873	**191**	C_16_H_17_O_9_	−0.6	Chlorogenic acid ^#^	C	[20]
8	4.28	389.1075	389.1084	**345**, 209, 183, 165	C_16_H_21_O_11_	−2.3	Secologanoside	A	[21,22]
9	4.32	209.0297	209.0297	**191**	C_6_H_9_O_8_	0.0	Mucic acid isomer	B	[20]
10	4.55	355.0659	355.0665	209, **191**	C_15_H_15_O_10_	−1.7	Caffeic acid glucuronide	B	[19]
11	4.62	951.0710	951.0740	**301**	C_41_H_27_O_27_	−3.2	Geraniin ^#^	C	[20]
12	4.63	885.2689	885.2665	739, 593, **285**	C_39_H_49_O_23_	2.7	Kaempferol-*O*-hexosyl-*O*-rhamnopyranoside-*O*-rhamnopyranoside-*O*-rhamnopyranoside	B	[23]
13	4.69	633.0724	633.0728	**463**	C_27_H_21_O_18_	−0.6	Corilagin ^#^	C	[20]
14	4.72	915.2777	915.2770	739, 593, **285**	C_40_H_51_O_24_	0.8	Methoxykaempferol-*O*-hexosyl-*O*-rhamnopyranoside-*O*-rhamnopyranoside-*O*-rhamnopyranoside	B	[24]
15	4.77	935.0778	935.0791	633, **301**	C_41_H_27_O_26_	−1.4	Galloyl-bis-HHDP-hexoside isomer	C	[25]
16	4.92	387.1651	387.1655	**225**	C_18_H_27_O_9_	−1.0	Tuberonic acid-*O*-hexoside	A	[17]
17	4.96	755.2014	755.2035	**609**, **301**	C_33_H_39_O_20_	−2.8	Quercetin rhamnosylrutinoside	B	[19]
18	4.97	935.0763	935.0791	633, **301**	C_41_H_27_O_26_	−3.0	Galloyl-bis-HHDPhexoside isomer	C	[25]
19	5.17	583.1667	583.1663	421, 259	C_26_H_31_O_15_	0.7	Unknown	B	-
20	5.29	739.2065	739.2086	593, **285**	C_33_H_39_O_19_	−2.8	Kaempferol-*O*-(α-L-rhamnosyl)-rutinoside	B	[26]
21	5.31	433.0818	433.0807	**301**	C_19_H_13_O_12_	2.5	Ellagic acid pentoside	C	[25]
22	5.32	179.0346	179.0344	**135**	C_9_H_7_O_4_	1.1	Caffeic acid ^#^	A	[15]
23	5.43	953.0896	953.0896	**301**, **169**	C_41_H_30_O_27_	0.0	Chebulagic acid	C	[27]
24	5.45	463.0863	463.0877	**301**, 300	C_21_H_19_O_12_	−3.0	Quercetin-3-*O*-galactose (hyperoside/hyperin)	A	[28]
25	5.50	593.1493	593.1506	**285**	C_27_H_29_O_15_	−2.2	Kaempferol-*O*-rutinoside	B	[26]
26	5.57	985.1158	985.1125	**300**, 169	C_42_H_33_O_28_	−3.3	Ellagic acid derivative	C	-
27	5.61	965.0858	965.0896	**300**, 169	C_42_H_29_O_27_	−3.8	Ellagic acid derivative	C	-
28	5.64	1035.3279	1035.3236	-	C_57_H_55_O_21_	4.0	Unknown	A	-
29	5.71	211.0601	211.0606	**167**	C_10_H_11_O_5_	−2.4	Eudesmic acid	B	[29]
30	5.79	300.9977	300.9984	-	C_14_H_5_O_8_	−2.3	Ellagic acid ^#^	C	[25]
31	5.89	537.1600	537.1608	-	C_25_H_29_O_13_	−1.5	Lippioside I	A	[17]
32	5.98	463.0871	463.0877	**301**	C_21_H_19_O_12_	−1.3	Quercetin-3-*O*-glucose (isoquercitrin) ^#^	C	[20]
33	6.17	623.1981	623.1976	461, **315**	C_29_H_35_O_15_	0.8	Verbascoside	A	[17]
34	6.26	421.1139	421.1135	**153**	C_20_H_21_O_10_	0.0	Amburoside A ^#^	B	[30]
35	6.43	447.0937	447.0927	**285**	C_21_H_19_O_11_	2.2	Kaempferol hexoside	C	[31]
36	6.65	315.0141	315.0141	-	C_15_H_7_O_8_	0.0	Methylellagic acid	C	-
37	6.68	451.1246	451.1240	**285**	C_21_H_23_O_11_	1.3	Catechin-*O*-hexoside	A	[32]
38	6.74	505.0981	505.0982	**301**	C_23_H_21_O_13_	−0.2	Quercetin acetylhexoside	A	[33]
39	6.78	451.1048	451.1029	**341**, 323, 217	C_24_H_19_O_9_	4.2	Cinchonain-Ib	C	[34]
40	6.96	463.1242	463.1242	**301**	C_22_H_23_O_11_	−0.4	Amburoside D	B	[35]
41	7.17	447.1281	447.1291	**285**	C_22_H_23_O_10_	−2.2	Isosakuranetin-hexoside	B	[36]
42	7.32	369.1552	369.1549	**165**	C_18_H_25_O_8_	0.8	Unknown	A	-
43	7.33	435.1280	435.1291	**273**, **167**	C_21_H_23_O_10_	−1.1	Amburoside B ^#^	B	[35,37]
44	7.37	435.1297	435.1291	**273**	C_21_H_23_O_10_	1.4	Phloretin-*O*-hexoside	A	[38]
45	7.59	613.1400	613.1405	**503**	C_26_H_29_O_17_	−0.8	Unknown	C	-
46	7.72	489.1039	489.1033	**285**	C_23_H_21_O_12_	1.2	Kaempferol-*O*-acetylhexoside	A	[39]
47	8.02	163.0396	163.0395	**119**	C_9_H_7_O_3_	0.6	Coumaric acid	B	[40]
48	8.26	193.0502	193.0501	**134**	C_10_H_9_O_4_	0.5	Ferulic acid ^#^	B	
49	8.40	285.0408	285.0399	**-**	C_15_H_9_O_6_	3.2	Kaempferol ^#^	A	[39]
50	9.48	271.0607	271.0606	**-**	C_15_H_11_O_5_	0.4	Naringenin ^#^	A	[41]
51	9.61	299.0552	299.0556	**284**	C_16_H_11_O_6_	−3.5	Kaempferide	A	[42]
52	9.81	353.1587	353.1600	**149**	C_18_H_25_O_7_	−3.7	Unknown	A	-
53	10.39	595.1808	595.1816	433, 283	C_31_H_31_O_12_	−1.3	Unknown	A	-
54	10.40	939.4971	939.4953	**921**, **877**, **551**, **455**	C_48_H_75_O_18_	1.9	Soyasaponin Be	B	[43]
55	10.80	909.4828	909.4848	**891**, **763**, **455**	C_47_H_73_O_17_	−2.0	Soyasaponin Bg	B	
56	11.64	1065.5343	1065.5329	**955**	C_47_H_85_O_26_	1.3	Unknown saponin	B	-
57	12.21	923.5007	923.5004	**-**	C_48_H_75_O_17_	0.3	Unknown saponin	B	-
58	12.55	925.5173	925.5161	**907**, **833**, **599**	C_48_H_77_O_17_	1.3	Hydroxyoleanenyl-deoxy-mannopyranosyl-hexopyranosyl-glucopyranosiduronic acid	B	[44]
59	12.89	893.4922	893.4899	**-**	C_48_H_77_O_17_	1.3	Unknown saponin	B	-
60	13.19	895.5078	895.5055	**-**	C_47_H_75_O_16_	2.6	Unknown saponin	B	-
61	13.50	1051.5511	747	**-**	C_54_H_83_O_20_	3.1	Unknown saponin	B	-
62	13.72	1049.5350	1049.5321	1019	C_54_H_81_O_20_	2.8	Unknown saponin	B	-
63	14.06	1021.5373	1021.5372	-	C_53_H_81_O_19_	0.1	Unknown saponin	B	-
64	14.45	1019.5243	1019.5216	-	C_53_H_79_O_19_	2.6	Unknown saponin	B	-

^a^ Peaks are numbered as indicated in the chromatograms shown in Figure 2a for *Lippia origanoides* (A), Figure 2b for *Amburana cearensis* (B), and Figure 2c for *Justicia pectoralis* (C). ^b^ Values of m/z marked in bold refer to diagnostic ions. ^#^ Identified with authentic analytical standards.

**Table 3 plants-14-03517-t003:** Comparative molecular docking results (Affinity and Energy) for the main α-glucosidase inhibitors identified from the extracts of *Lippia origanoides*, *Amburana cearensis* and *Justicia pectoralis*.

Compounds	CID	Score	Intermolecular Energy	vdW Energy	Electrostatic Energy
Acarbose	9811704	−7.465	−75.493	−9.318	−66.175
Quercetin-3-*O*-galactose (hyperoside)	5281643	−7.994	−44.180	−16.492	−27.688
Quercetin-3-*O*-glucose (isoquercitrin)	25203368	−7.514	−41.106	−20.638	−20.468
Trimethoxy benzoic acid (eudesmic acid)	8357	−6.391	−39.873	2.921	−42.794

## Data Availability

The data presented in this study are available on request from the corresponding author. The data are not publicly available due to institucional restrictions.

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
