# Peer review of "Searching for Hypoglycemic Compounds from Brazilian Medicinal Plants Through UPLC-HRMS and Molecular Docking"

_plants, 2025, doi:10.3390/plants14223517_

Round 1
Reviewer 1 Report
Comments and Suggestions for Authors
The manuscript entitled ‘Searching for α-Glucosidase Inhibitors From Brazilian Medicinal Plants Through UPLC-HRMS and Molecular Docking' is very interesting.
The introduction is well written, however, it lacks information about the glucosidase pathways in people with diabetes and about the plants used in the present study, including whether they are angiosperms, monocotyledonous, or dicotyledonous... Adding photos of these species (Material and Methods) would provide more information and improve the paper.
Figure 3: Please, add to the caption the website/software used to construct the bind between compound and glucosidase.
Line 196: ‘inhibit in vitro and in vivo α-glucosidase activities’. Please correct this, since ‘in vitro’ is written twice.
Line 224: Sometimes ‘in vitro’ is in italic, and sometimes not. Please, standardize this.
Line 231: Please, add the period in the sentence.
Line 301: Why did you test 0.312 and 10 mg. mL-1? Please, explain these concentrations.
Line 312: What normality and homogeneity tests were performed?
Did you perform spectrophotometric screening of each extract to identify the absorption wavelengths? This would help corroborate the UPLC-HRMS findings.
Because of this, I recommend accepting the paper with major revision.
Author Response
The manuscript entitled ‘Searching for α-Glucosidase Inhibitors From Brazilian Medicinal Plants Through UPLC-HRMS and Molecular Docking' is very interesting.
The introduction is well written, however, it lacks information about the glucosidase pathways in people with diabetes
Response: Yes, thank you for bringing this to our attention. To clarify the introduction, we have inserted the following sentences:
“Under diabetic conditions, the activity of these enzymes is markedly elevated, contributing to postprandial hyperglycemia. Located on the brush border of the intestinal epithelium, they catalyze glucose release from oligosaccharides and disaccharides, thereby accelerating its absorption. Studies have shown that insulin deficiency abnormally increases disaccharidase activity in animal models (Liu et al., 2011) and that patients with T2DM exhibit approximately 1.5-fold higher α-glucosidase expression than healthy individuals (Dirir et al., 2022). This enhanced enzymatic activity intensifies intestinal glucose uptake and exacerbates metabolic imbalance, reinforcing the therapeutic relevance of α-glucosidase inhibitors such as acarbose and miglitol to control postprandial blood glucose levels (Hossain et al., 2020).”
Reference:
- Liu L. et al. Insulin deficiency induces abnormal increase in intestinal disaccharidases. Biochemical and Biophysical Research Communications, 2011, 414(3): 588–593.
- Dirir A. M. et al. A review of α-glucosidase inhibitors from plants as potential candidates for the treatment of type 2 diabetes. Phytochemistry Reviews, 2022, 21: 1049–1079.
- Hossain U. et al. An overview on the role of bioactive α-glucosidase inhibitors in controlling postprandial hyperglycemia. Journal of Diabetes Research, 2020, Article ID 7480666.
…about the plants used in the present study, including whether they are angiosperms, monocotyledonous, or dicotyledonous...
Response: Thank you for this observation. We have clarified this question by adding the following sentence: “All species studied are dicotyledons that occur across many countries of South and Central America, except L. ferrea, which is endemic to Brazil.”
Adding photos of these species (Material and Methods) would provide more information and improve the paper.
Response: Thank you for bringing this observation. As requested, photos of the three bioactive plants have been added to the manuscript (figure 4) .
Figure 3: Please, add to the caption the website/software used to construct the bind between compound and glucosidase.
Response:The information of the software LIGPLOT+ has been included in Figure 3 caption.
Line 196: ‘inhibit in vitro and in vivo α-glucosidase activities’. Please correct this, since ‘in vitro’ is written twice.
Response: Thank you for bringing this observation. The sentence was kept. Actually, it refers to activities measured by in vitro and in vivo assays.
Line 224: Sometimes ‘in vitro’ is in italic, and sometimes not. Please, standardize this.
Response: Thank you for bringing this observation. All words “in vitro” have been italicized.
Line 231: Please, add the period in the sentence.
Response: Thank you for bringing this observation. A period has been added.
Line 301: Why did you test 0.312 and 10 mg. mL-1? Please, explain these concentrations.
Response: Excellent question, thank you for asking. We'll add this explanation to the text. "The concentrations of 0.312 and 10 mg mL⁻¹ were selected based on preliminary testing, restricted to alpha-glucosidase, performed to determine the effective concentration range of the compounds. The objective was to encompass both the minimum concentration capable of producing a detectable effect and the maximum concentration feasible for analysis. Thus, these concentrations represent the extremes of a previously established experimental range to evaluate the biological response in a broad and comparative manner."
Line 312: What normality and homogeneity tests were performed?
Response: We thank the reviewer for raising this important point. In response, we have now included additional details on the normality and homogeneity tests performed on the data to ensure the assumptions for ANOVA were met. Below are the details:
Normality Test:
To verify whether the data followed a normal distribution (a fundamental assumption for performing ANOVA), we conducted the Shapiro-Wilk test. The results indicated that there was no significant deviation from normality for any of the groups, as the p-values were greater than 0.05 (p > 0.05). This suggests that, based on the Shapiro-Wilk test, there is insufficient evidence to reject the normality assumption. Additionally, we applied the Kolmogorov-Smirnov test, when necessary, which corroborated the results from the Shapiro-Wilk test. Both tests confirmed that the data from all groups can be considered normally distributed.
Homogeneity of Variances:
Another important assumption for ANOVA is homogeneity of variances, which means that the variances of the different groups should be similar. To test this assumption, we applied the Brown-Forsythe test, which is considered robust even when the data do not follow a normal distribution. The Brown-Forsythe test yielded F(7,26) = 1.803, with a p-value of 0.1293, which is greater than 0.05. This indicates that there are no significant differences in variances between the groups. In other words, the variances of the groups are homogeneous, validating this assumption of ANOVA.
Thus, we have added the text below to the Methods section (Statistical Analyses) to clarify the statistical issues and tests performed to ensure the validity of our ANOVA analysis:
4.7 Statistical Analyses
All data were analyzed using GraphPad Prism 9 (GraphPad Software, La Jolla, CA, USA). The normality of the data was assessed using the Shapiro-Wilk test for each experimental group. Data were considered normally distributed when p-values were greater than 0.05. Additionally, the Kolmogorov-Smirnov test was applied when necessary to further confirm the results from the Shapiro-Wilk test. To evaluate the homogeneity of variances among groups, the Brown-Forsythe test was used, which is robust even when data do not follow a normal distribution. Homogeneity of variances was considered met when the p-value was greater than 0.05. A one-way analysis of variance (ANOVA) was conducted to compare the effects of treatments relative to controls. When appropriate, Dunnett’s post hoc test was used for multiple comparisons between groups, with differences considered significant at p < 0.05. Statistical significance was set at α = 0.05 for all analyses. Results are presented as means ± Standard Error of the Mean (SEM) from two independent experiments (N = 2), with each experiment performed in technical triplicates (n = 3).
Did you perform spectrophotometric screening of each extract to identify the absorption wavelengths? This would help corroborate the UPLC-HRMS findings.
Because of this, I recommend accepting the paper with major revision.
Response: We thank the reviewer for this thoughtful and constructive comment. However, we respectfully disagree that spectrophotometric screening would provide additional or complementary information to corroborate the UPLC-HRMS results. Given the high chemical complexity of the crude extracts—comprising numerous metabolites from distinct chemical classes—and the low specificity of UV–Vis absorption bands, it would be difficult to attribute specific absorption maxima to individual components tentatively identified by UPLC-HRMS. Therefore, we consider that such an analysis would not substantially strengthen the chemical characterization presented in this study.

Reviewer 2 Report
Comments and Suggestions for Authors
This is a well-structured and technically sound pharmacognosy study that integrates enzyme inhibition assays, UPLC-HRMS-based chemical profiling, and molecular docking. The study’s novelty lies in identifying Lippia origanoides, Amburana cearensis, and Justicia pectoralis as potential sources of α-glucosidase inhibitors. However, several aspects could be refined to enhance scientific rigor, reproducibility, and interpretability. To the best of my knowledge, I am summarizing below points for the improvement of manuscript-
- The research article title should be concise, descriptive, and informative. The title “Searching for α-Glucosidase Inhibitors…” is somewhat vague. I would suggest reconsidering changing the title.
- Abstract is slightly long. Kindly rewrite following journal guidelines.
- The introduction is comprehensive but mostly descriptive. It does not clearly define the hypothesis or the knowledge gap. Explicitly state the rationale of the study.
- Clarify why aqueous extracts were chosen over hydroalcoholic or methanolic ones, as this influences compound recovery and enzyme interaction outcomes.
- The α-glucosidase inhibition assay and ex-vivo disaccharidase assays are standard but lack details on positive/negative controls, replicates, and statistical robustness (e.g., n-values for each assay type). Please Include the number of biological replicates (not just technical repeats) and specify if experiments were performed in triplicate across independent batches.
- The extraction conditions (100 °C for 5 min) are likely to degrade thermolabile metabolites (e.g., glycosides). Consider justifying the choice of decoction and compare to literature standards.
- In Docking studies, Include re-docking RMSD values to validate the docking protocol. Explain how grid box dimensions were selected relative to the active site.
- Also, if possible, provide visualization of hydrogen-bonding residues and hydrophobic pockets for top compounds.
- The enzyme inhibition results are expressed as % inhibition and IC₅₀, but dose–response curves are missing. These are essential for validating IC₅₀ I would suggest to include full dose–response plots with nonlinear regression fits in Supplementary Materials.
- The comparison with acarbose should include statistical significance testing (e.g., p < 0.05), not just descriptive comparisons.
- Table 2 is exhaustive but lacks fragmentation spectra or MS/MS confirmation for key peaks. Tentative identifications should be supported with m/z error < 5 ppm, diagnostic ions, and literature references (some already cited, but more justification is needed).
- The docking discussion is strong but speculative. Complement in-silico results with a ligand efficiency (LE) or binding energy normalization per atom, and discuss drug-likeness (Lipinski’s rule, logP) to better connect docking results to pharmacological potential.
- Figure 3 (docking visualization) should include binding site residues labels (e.g., Asp, Glu, Arg) and a scale bar for clarity.
- The discussion cites prior studies but could benefit from critical comparison — for example, discuss differences in solvent system, plant part, or assay model leading to variation in reported potency.
- The claim that “hyperoside surpasses acarbose in binding predictions” is based solely on docking and should be softened.
- In Figure 1 Axis labels too small; “specific activity” units are unclear.
- In references some citations are outdated or missing DOIs. Ensure metabolomics and docking references are included.
- In Introduction section, line 43, cite the reference https://doi.org/10.1080/14786419.2019.1633650
- If possible, include LC–MS/MS chromatograms in Supplementary Information.
- Add positive control phytochemical (e.g., quercetin standard) to compare inhibition potency.
- Include PCA or heatmap visualization for compound classes among extracts would strengthen chemometric interpretation.
- I would also suggest conducting cytotoxicity screening to assess therapeutic window.
Author Response
The research article title should be concise, descriptive, and informative. The title “Searching for α-Glucosidase Inhibitors…” is somewhat vague. I would suggest reconsidering changing the title.
Response:Yes, thank you for bringing this to our attention. We have replaced the term “α-Glucosidase Inhibitors” by “hypoglycemic compounds”. Thus, the new title is ‘Searching for Hypoglycemic Compounds from Brazilian Medicinal Plants Through UPLC-HRMS and Molecular Docking’
Abstract is slightly long. Kindly rewrite following journal guidelines.
Response: We thank the reviewer for this helpful suggestion. The abstract has been revised and shortened from 197 words to 165 words. In addition to reducing the length, the revised version was refined to include key new findings resulting from the revision process, specifically: 1- the in vitro α-glucosidase inhibition assay of isoquercitrin, 2-toxicological assessment in zebrafish, and in silico ADME analysis of the main active compounds. These additions were concisely integrated to preserve clarity and ensure that the abstract accurately reflects the updated experimental scope of the study.
The introduction is comprehensive but mostly descriptive. It does not clearly define the hypothesis or the knowledge gap. Explicitly state the rationale of the study.
Response: We thank the reviewer for this valuable suggestion. In response, we have added the following sentence to the Introduction to better define the knowledge gap and hypothesis: “Despite numerous reports describing enzyme inhibition by plant-derived compounds, most studies have focused on species traditionally used for diabetes treatment. However, several tropical plants with well-documented medicinal properties remain chemically and pharmacologically underexplored for this purpose. We hypothesize that their flavonoids and phenolic acids-rich extracts may be interesting sources of antidiabetic compounds by modulating carbohydrate metabolism through enzyme inhibition and complementary antioxidant mechanisms.”
Clarify why aqueous extracts were chosen over hydroalcoholic or methanolic ones, as this influences compound recovery and enzyme interaction outcomes.
Response: We thank the reviewer for this pertinent observation.
The choice of aqueous extracts was motivated by their alignment with pharmacopoeial preparations—particularly infusions—recognized for their safety, simplicity, and traditional use in ethnopharmacological contexts. Moreover, aqueous extraction mimics the conditions under which these plants are commonly consumed, providing biological relevance for evaluating enzyme inhibition under physiologically compatible conditions. While hydroalcoholic or methanolic solvents can enhance compound recovery, the aqueous medium offers a safer and more translationally meaningful extract for potential therapeutic applications.
The α-glucosidase inhibition assay and ex-vivo disaccharidase assays are standard but lack details on positive/negative controls, replicates, and statistical robustness (e.g., n-values for each assay type). Please Include the number of biological replicates (not just technical repeats) and specify if experiments were performed in triplicate across independent batches.
Thank you for your insightful comment. We fully agree that specifying the positive/negative controls, biological replicates, and statistical approach is essential to ensure experimental robustness and reproducibility.
In the disaccharidase assay (Figure 1), although it was initially stated that 7 animals were used per group, in fact, 7 animals were euthanized for the entire study. From each animal, two intestinal segments were collected and allocated across the experimental groups. For each group, n = 2 biological replicates, and all readings were conducted in technical triplicate. This clarification has been updated in the figure legend. In the α-glucosidase inhibition assay (Table 1), experiments were performed with n = 2 biological replicates, each measured in technical triplicate. Acarbose was employed as the positive control in both assays (disaccharidase and α-glucosidase). Additionally, isoquercitrin was included as a secondary positive control for the α-glucosidase assay, as shown in the supplementary data. For the negative controls, either the absence of the test substance or the vehicle alone was used, respectively, in both assays. Statistical analysis was carried out using one-way ANOVA followed by multiple comparison tests, with p < 0.05 considered statistically significant. These details have been incorporated into the revised manuscript to enhance clarity and methodological transparency.
- The extraction conditions (100 °C for 5 min) are likely to degrade thermolabile metabolites (e.g., glycosides). Consider justifying the choice of decoction and compare to literature standards.
We appreciate the reviewer’s comment. There was a misunderstanding in the original description. The extracts were actually prepared by infusion, not decoction, following the Brazilian Pharmacopeia guidelines for preserving thermolabile constituents. The text has been corrected and the reference added accordingly.
In Docking studies, Include re-docking RMSD values to validate the docking protocol. Explain how grid box dimensions were selected relative to the active site.
Response:We appreciate the reviewers' comments. To validate the accuracy and reliability of our molecular docking protocol, we performed a re-docking procedure. The native co-crystallized ligand (Acarbose and 2QMJ) was extracted from the protein's active site and then re-docked back into the same binding site using the defined parameters.
The Root-Mean-Square Deviation (RMSD) between the original and the re-docked ensemble was obtained as 0.9Å (for the protein) and 1.9Å (for the ligand), respectively. An RMSD value of ≤ 2.0 Å is widely accepted in the literature as indicative of a successful and reproducible docking protocol.
Figure R1: RMSD calculation for 2QMJ original structure and human α-glucosidase with hyperoside (quercetin-3-O-galactose) from Lippia origanoides extract.
The binding site of 2QMJ is a deep, well-defined catalytic cleft within a TIM barrel, utilizing a combination of catalytic aspartates, a network of hydrogen bonds, and key aromatic residues to tightly bind the inhibitor acarbose. To encompass these regions, we defined a grid box centered at coordinates x = 55.41 Å, y = 98.61 Å, and z = 20.59 Å, with a grid spacing of 0.42 Å. This box was sized to include both the primary binding site and the adjacent cleft lined with aromatic and polar residues.
Also, if possible, provide visualization of hydrogen-bonding residues and hydrophobic pockets for top compounds.
Response: We agree with the recommendation and provided the pictures of binding attributes for the ligand in the binding site of the alpha glucosidase shown in the new depicted picture in new Figure 3. The residues ARG283, ASP777 and GLU114 were labeled in the 3D image (A), while in the Ligplot image (B) are labeled in green. Hydrophobic residues are displayed in gray solid surface (GLY282, HIS115, ALA780, ILE523, LYS534, LYS776, ALA509, PHE535, HIS645 and ALA285) (A), and displayed in red arcs in (B).
Figure 3: Molecular interaction of Hyperoside (Quercetin-3-O-galactose) with the human α-glucosidase active site. (A) Three-dimensional representation of the docked pose. The protein surface is shown as a transparent dot field. Hyperoside is depicted as ball-and-stick. Residues forming hydrogen bonds are shown as ball-and-stick and labeled; residues involved in hydrophobic interactions are highlighted with a gray solid surface. (B) LIGPLOT two-dimensional ligand-protein interaction diagram. Hydrogen bonds are shown as green dashed lines, and hydrophobic contacts are indicated by red arcs.
The enzyme inhibition results are expressed as % inhibition and IC₅₀, but dose–response curves are missing. These are essential for validating IC₅₀ I would suggest to include full dose–response plots with nonlinear regression fits in Supplementary Materials.
We appreciate the reviewer’s valuable suggestion. The complete dose–response curves with nonlinear regression fits have now been included in the Supplementary Materials (Figure S1). These plots were generated using GraphPad Prism 9 to calculate IC₅₀ values and confirm the reliability of the inhibition data.
The comparison with acarbose should include statistical significance testing (e.g., p < 0.05), not just descriptive comparisons.
We thank the reviewer for this helpful suggestion. We have now performed statistical analyses to compare the inhibitory activity of the tested samples with acarbose. The results were analyzed using a one-way ANOVA followed by Dunnett’s post hoc test (p < 0.05), and statistically significant differences are now indicated in Table S1. Significant differences were observed between Justicia pectoralis (p = 0.0023) and Amburana cearensis (p = 0.0276) compared to acarbose, but not for Lippia origanoides (p = 0.2928). This statistical analysis strengthens the comparison with the reference inhibitor.
- Table 2 is exhaustive but lacks fragmentation spectra or MS/MS confirmation for key Tentative identifications should be supported with m/zerror < 5 ppm, diagnostic ions, and literature references (some already cited, but more justification is needed).
We thank the reviewer for this insightful comment. The tentative identification of metabolites was carefully performed considering all parameters mentioned, including accurate mass with an error < 5 ppm, diagnostic fragment ions, and literature references, as detailed in Table 2. Additionally, compound annotation was supported by previous reports of these metabolites in the same species, genus, or botanical families, based on a comprehensive taxonomic survey using natural product databases (e.g., SciFinder). To enhance clarity, the column previously labeled “Product ions” was replaced with “MS/MS fragment ions,” and the corresponding diagnostic ions were highlighted in bold. Furthermore, authentic standards were employed to corroborate the identification whenever available.
The docking discussion is strong but speculative. Complement in-silico results with a ligand efficiency (LE) or binding energy normalization per atom, and discuss drug-likeness (Lipinski’s rule, logP) to better connect docking results to pharmacological potential.
Response: We thank the reviewer for the insightful comment. Thus, we have provided the ADME prediction for the molecular docking ligands and detailed results are in the supplementary material.
In the methods section we included:
“ADME Properties Prediction
The pharmacokinetic profiles—encompassing Absorption, Distribution, Metabolism, and Excretion (ADME)—were predicted using the SwissADME web tool (http://www.swissadme.ch). The smiles files were uploaded directly to the server. The analyses were run using the default parameters for all compounds.”
In Results section, we have included the following sentence:
The ADME and drug-likeness predictions support the pharmacological interpretation of the docking results by highlighting differences in molecular properties that influence absorption and systemic availability. Hyperoside (quercetin-3-O-galactoside) and isoquercitrin (quercetin-3-O-glucoside) displayed physicochemical profiles typical of natural product glycosides, including high polarity and an elevated number of hydrogen bond donors and acceptors (Santos et al., 2025). These characteristics led to multiple Lipinski’s rule of five violations, resulting in low predicted gastrointestinal (GI) absorption and poor blood–brain barrier permeability. Such profiles are consistent with their limited oral bioavailability but may still allow local intestinal enzyme inhibition, which aligns with their observed α-glucosidase and disaccharidase inhibitory effects. In contrast, trimethoxybenzoic acid exhibited a favorable drug-likeness profile, showing no Lipinski’s violations, good oral bioavailability (bioavailability score = 0.85), high predicted water solubility, and potential blood–brain barrier permeability. These properties suggest that this compound could achieve systemic exposure and potentially exert both central and peripheral pharmacological effects. Meanwhile, acarbose—although a clinically established α-glucosidase inhibitor—violates three of Lipinski’s rules, consistent with its poor intestinal absorption and low oral bioavailability. Despite this, its therapeutic efficacy relies on localized gastrointestinal action, which parallels the mechanism proposed for quercetin glycosides. Importantly, none of the analyzed compounds were predicted to exhibit cytochrome P450 inhibitory liabilities, minimizing the likelihood of drug–drug interactions. However, their relatively high synthetic accessibility scores, particularly for acarbose, indicate substantial structural complexity (see Supplementary Material).
Figure 3 (docking visualization) should include binding site residues labels (e.g., Asp, Glu, Arg) and a scale bar for clarity.
Response: we appreciate the suggestion and Figure 3 was edited to display the amino acid residues involved in H-bond and hydrophobic interactions with the ligand. H-bond residues are labeled and displayed in ball & stick while hydrophobic interactions residues are displayed in gray solid surface to allow the positioning of the cleft.
The discussion cites prior studies but could benefit from critical comparison — for example, discuss differences in solvent system, plant part, or assay model leading to variation in reported potency.
We appreciate the reviewer’s valuable suggestion. To address this point without diverting the main focus of the discussion, we have added a clarifying statement highlighting that variations in reported bioactivity may result from methodological differences among studies. The following sentence was inserted into the discussion: This difference of chemical composition between the extracts might be due to part plant used and the extraction method (e.g. solvent type).
The claim that “hyperoside surpasses acarbose in binding predictions” is based solely on docking and should be softened.
We appreciate the reviewer’s careful observation. The sentence has been revised to avoid overstatement, clarifying that the conclusion derives exclusively from in silico simulations: “Molecular docking identified hyperoside as a compound with strong predicted affinity for α-glucosidase, comparable to acarbose, suggesting its potential contribution to the observed inhibitory activity.”
- In Figure 1 Axis labels too small; “specific activity” units are unclear.
Response: We appreciate the reviewer’s observation. In the revised version, we have increased the font size of all axis labels in Figure 1 to improve readability. In addition, we have clarified the units of “specific activity” in both the figure legend and the Materials and Methods section, now expressed as U mg⁻¹ protein.
In references some citations are outdated or missing DOIs.
We thank the reviewer for this observation. All references have been carefully revised, and outdated citations were replaced with more recent and relevant studies. Additionally, missing DOIs have been verified and added where available to ensure accuracy and completeness of the reference list.
- Ensure metabolomics and docking references are included.
We have added the following sentence in Introduction with its respective reference: “UPLC-HRMS has emerged as a key analytical platform for metabolomic approaches aiming at the rapid characterization of bioactive compounds in plant extracts. UPLC-HRMS combines efficient chromatographic separation with high mass accuracy, enabling the simultaneous identification of multiple metabolites in complex plant matrices. The exact mass data allow reliable determination of molecular formulas and dereplication of known bioactive compounds. By generating detailed chemical fingerprints, this technique accelerates phytochemical screening and identification of biologically relevant metabolites, promoting a more efficient exploration of natural products (Meunier et al., 2024). Ref.: Meunier, M.; Schinkovitz, A.; Derbré, S. Current and emerging tools and strategies for the identification of bioactive natural products in complex mixtures. Nat. Prod. Rep., 2024, 41, 1766.
- In Introduction section, line 43, cite the reference https://doi.org/10.1080/14786419.2019.1633650.
We thank the reviewer for suggesting this reference. We carefully reviewed the cited article (DOI: 10.1080/14786419.2019.1633650). However, it reports the isolation of a compound from Putranjiva roxburghii without addressing aspects related to glucose metabolism, insulin resistance, or oxidative stress—topics central to the present work. Therefore, we respectfully consider that including this citation would not contribute relevant context to the Introduction. Nonetheless, we have ensured that all key and recent literature directly related to T2DM pathophysiology is properly cited.
- If possible, include LC–MS/MS chromatograms in Supplementary Information.
The LC–MS/MS chromatograms have been shown on Table 2.
Add positive control phytochemical (e.g., quercetin standard) to compare inhibition potency.
We appreciate the reviewer’s insightful suggestion. In response, we conducted an additional in vitro α-glucosidase inhibition assay using an authentic standard of isoquercitrin (quercetin-3-O-glucose), which had been identified through molecular docking as the main compound responsible for the enzyme inhibitory activity observed in the Justicia pectoralis extract. The experimental results confirmed the prediction, as isoquercitrin exhibited a lower IC₅₀ value than acarbose, demonstrating higher inhibitory potency. The corresponding dose–response curve is provided in the Supplementary Material (Figure S4).
- Include PCA or heatmap visualization for compound classes among extracts would strengthen chemometric
We thank the reviewer for this valuable suggestion. However, we believe that PCA or heatmap analyses would not provide additional insights relevant to the main objective of our study. As the investigated species belong to distinct botanical families, substantial chemical variability among their extracts is expected. Therefore, multivariate analyses such as PCA or heatmaps would primarily reflect taxonomic divergence rather than contribute meaningfully to the interpretation of bioactive compound patterns.
I would also suggest conducting cytotoxicity screening to assess therapeutic window.
Response: We appreciate the reviewer’s valuable suggestion. In response, we have conducted toxicity assays for the three bioactive extracts (Lippia origanoides, Amburana cearensis, and Justicia pectoralis) using acute toxicity 96 h zebra fish model. The results are now included in the revised manuscript in subsection 2.2 as can be seen below. Additional information can be found in supplementary material.
2.2. Toxicological Evaluation of the Bioactive Extracts.
The toxicological assessment of the bioactive extracts in adult zebrafish (Danio rerio) indicated low acute toxicity for L. origanoides and A. cearensis extracts, with negligible mortality observed up to the highest tested concentration (100 mg mL⁻¹) (Table S1- supplementary material). The LC₅₀ values for both extracts were estimated to exceed this limit, supporting their classification as low-hazard substances according to the OECD Test Guideline 203 and GHS/CLP criteria. In contrast, J. pectoralis extract produced limited mortality at 50 mg mL⁻¹, yielding an estimated LC₅₀ above this concentration but below 100 mg mL⁻¹. Consequently, it can be provisionally classified within GHS Category 3 (LC₅₀ between 10 and 100 mg mL⁻¹), pending further assays to refine its toxicological profile.
Furthermore, the description of the experiment has been provided in Material& Methods section:
For toxicity assay, adult wild-type zebrafish (Danio rerio) of both sexes were obtained from a local supplier (Agroquímica Produtos Veterinários, Fortaleza, CE, Brazil). The specimens were approximately 60–90 days old, measuring 3.5 ± 0.5 cm in length and weighing 0.4 ± 0.1 g. Upon arrival, fish were acclimated for 24 h in rectangular glass tanks (30 × 15 × 20 cm) containing dechlorinated water (ProtecPlus®) maintained at 25 °C and neutral pH, and equipped with air pumps and submerged filters, as described by Magalhães et al. [reference]. Zebrafish were maintained under a 14:10 h light/dark photoperiod and fed ad libitum until 24 h prior to experimentation. All experimental procedures were approved by the Animal Ethics Committee of the Federal University of Ceará (CEUA-UFC; protocol no. 2101202201/2025).

Round 2
Reviewer 1 Report
Comments and Suggestions for Authors
All the modifications were made, and this resulted in a very interesting and complete manuscript.
I accept the paper in present form.